# Spatial control of lipid droplet proteins by the ERAD ubiquitin ligase Doa10

Annamaria Ruggiano[1,2,†,‡], Gabriel Mora[1,2,‡], Laura Buxó[1,2] & Pedro Carvalho[1,2,*]

## Abstract

The endoplasmic reticulum (ER) plays a central role in the biogenesis of most membrane proteins. Among these are proteins localized to the surface of lipid droplets (LDs), fat storage organelles delimited by a phospholipid monolayer. The LD monolayer is often continuous with the membrane of the ER allowing certain membrane proteins to diffuse between the two organelles. In these connected organelles, how some proteins concentrate specifically at the surface of LDs is not known. Here, we show that the ERAD ubiquitin ligase Doa10 controls the levels of some LD proteins. Their degradation is dependent on the localization to the ER and appears independent of the folding state. Moreover, we show that by degrading the ER pool of these LD proteins, ERAD contributes to restrict their localization to LDs. The signals for LD targeting and Doa10-mediated degradation overlap, indicating that these are competing events. This spatial control of protein localization is a novel function of ERAD that might contribute to generate functional diversity in a continuous membrane system.

**Keywords** Doa10; endoplasmic reticulum; ERAD; lipid droplets; protein degradation

**Subject Categories** Membrane & Intracellular Transport; Protein Biosynthesis & Quality Control

**The EMBO Journal (2016) 35: 1644–1655**

## Introduction

The endoplasmic reticulum (ER) plays a central role in the biogenesis of membrane and secretory proteins, facilitating the folding and the post-translational modifications necessary for their function (Braakman & Hebert, 2013). Protein folding in the ER is under the surveillance of stringent quality control and polypeptides failing to acquire the native structure are eliminated by ER-associated degradation (or ERAD) (Smith *et al*, 2011; Christianson & Ye, 2014; Ruggiano *et al*, 2014). This process involves the recognition of a substrate, its ubiquitination by an ER ubiquitin ligase, membrane

extraction facilitated by the cytoplasmic Cdc48 ATPase, and delivery to the proteasome for degradation. These events are coordinated by ER membrane-embedded protein complexes that have at their core an ubiquitin ligase. While highly conserved across eukaryotes, the mechanisms of ERAD are better characterized in the yeast *S. cerevisiae*. In yeast, genetic and biochemical studies identified three ubiquitin ligase complexes involved in ERAD, the Hrd1, Doa10, and Asi complexes, showing different specificity for misfolded proteins (Hampton *et al*, 1996; Swanson *et al*, 2001; Carvalho *et al*, 2006; Denic *et al*, 2006; Gauss *et al*, 2006; Foresti *et al*, 2014; Khmelinskii *et al*, 2014). Besides misfolded proteins, Hrd1 and Doa10 complexes were shown to degrade some folded, functional proteins but in a regulated manner, only upon a specific signal. Regulated degradation is important to control certain ER functions, such as sterol biosynthesis (Ruggiano *et al*, 2014). The Asi complex localizes specifically to the inner nuclear membrane (INM), preventing the accumulation of misfolded proteins in this highly specialized ER domain (Boban *et al*, 2006; Foresti *et al*, 2014; Khmelinskii *et al*, 2014). Moreover, the Asi complex degrades ER proteins mistargeted to the INM, suggesting that ERAD also integrates spatial cues.

The ER also has a direct role in the biogenesis of other organelles, such as lipid droplets (LDs) (Thiam *et al*, 2013; Pol *et al*, 2014). These storage organelles consist of a core of neutral lipids, mainly triacylglycerides (TAG) and sterol esters, enclosed by a phospholipid monolayer and a set of LD-specific proteins, primarily enzymes promoting the synthesis, remodeling, and consumption of lipids in LDs. Therefore, the metabolic status of individual LDs is largely determined by the proteins at their surface. Typical membrane proteins with hydrophilic domains on both sides of the bilayer are not favorably accommodated in the monolayer of LDs; the association of proteins with the surface of these organelles is instead mediated by either amphipathic helices or hydrophobic hairpins (Thiam *et al*, 2013). Proteins with the former motif are recruited to LDs directly from the cytoplasm. In contrast, proteins of the latter type are initially targeted and membrane-inserted in the ER, and subsequently targeted to the LD monolayer, which in yeast and in a large fraction of mammalian LDs is continuous with the outer leaflet of the ER membrane (Jacquier *et al*, 2011; Wilfling *et al*, 2013). How, among all the ER membrane proteins containing hydrophobic hairpins, some concentrate specifically at the LD monolayer is not

1  Centre for Genomic Regulation (CRG), Barcelona Institute of Science and Technology, Barcelona, Spain
2  Universitat Pompeu Fabra (UPF), Barcelona, Spain
   *Corresponding author. Tel: +34 933 160 286; E-mail: pedro.carvalho@crg.eu
   ‡These authors contributed equally to this work
   †Present address: Department of Molecular Cell Biology, Max Planck Institute of Biochemistry, Martinsried, Germany

 

entirely clear. In a few cases, positively charged amino acids flanking the hairpin favor their retention in LDs; however, a consensus signal or sequence has not been identified (Ingelmo-Torres *et al*, 2009). Moreover, it is unclear why some proteins concentrate in LDs soon after their integration at the ER, while others accumulate slower and, in some cases, depending on the metabolic state of the cells. For example, in quiescent yeast cells the enzyme Dga1 localizes to LDs where it synthesizes TAG, while in dividing cells a prominent fraction of Dga1 localizes to the ER where it is less active (Oelkers *et al*, 2002; Sorger & Daum, 2002; Jacquier *et al*, 2011; Markgraf *et al*, 2014).

Here, we identified a subset of LD proteins as substrates of the ERAD ubiquitin ligase Doa10. We show that ERAD targets specifically the ER pool of these proteins. The common feature among these Doa10 clients is the presence of a hydrophobic hairpin involved in LD targeting. We show that the signals for Doa10-mediated degradation and for LD targeting overlap, indicating that these are competing events and a potential target for regulation. Altogether, our data indicate that ERAD-mediated degradation of a subset of LD proteins in the ER restricts their localization to the LD surface. We propose a novel function of ERAD in protein spatial control that contributes to organelle identity by limiting the accumulation of LD-specific proteins in the ER.

## Results

### ERAD degrades LD proteins

Quantitative proteomic screenings recently performed in our laboratory generated a long list of potential endogenous ERAD substrates (Foresti *et al*, 2013, 2014). Among these, the LD proteins Pgc1, Dga1, and Yeh1 were over-represented in the *doa10Δ* mutant in comparison with wt cells, suggesting that their abundance might be controlled by ERAD. Curiously, *doa10Δ* cells also have defects in LD morphology strengthening a potential connection between ERAD and LD regulation (Fei *et al*, 2008, 2009). The levels of other LD proteins, such as Erg6, Pet10, Osw5, or Hfd1, were unaffected in *doa10Δ* cells, as detected by SILAC and cycloheximide chase experiments. Here, we focused on Pgc1 to characterize the Doa10-mediated degradation of certain LD proteins. To directly assess the role of Doa10 in controlling Pgc1 levels, we performed cycloheximide chase experiments. In wt cells, endogenously expressed Pgc1 was short-lived (half-life of ~45 min). In agreement with the proteomics data, its degradation was significantly delayed in cells lacking the ubiquitin ligase Doa10 or its binding partners Ubc6 and Ubc7, but was not affected in *hrd1Δ* cells, lacking another ERAD ubiquitin ligase (Fig 1A). To further characterize Pgc1 degradation, we analyzed its ubiquitination. While in wt cells ubiquitin-conjugated 3HA-Pgc1 was readily detected, Pgc1 ubiquitination was reduced in cells lacking Doa10 (Fig 1B). Although decreased, neither the degradation nor the ubiquitination of Pgc1 was blocked in *doa10Δ* cells, prompting us to search for additional ubiquitin ligases involved in its degradation. While deletion of all three ERAD ubiquitin ligases in *doa10Δ hrd1Δ asi1Δ* cells did not further stabilize Pgc1 (Fig EV1A), the soluble ubiquitin ligase Ubr1 was redundant with Doa10 as Pgc1 was more stable in *doa10Δ ubr1Δ* mutant (Fig EV1B). In fact, in *ubr1Δ* cells Pgc1 degradation was slowed down, even if not to the same extent as in *doa10Δ* mutant. Similar results were obtained for the LD proteins Dga1 and Yeh1, also

identified in the SILAC dataset (Fig EV2). In agreement with these findings, Ubr1 was shown to be redundant with Doa10 in the degradation of other membrane-bound substrates (Stolz *et al*, 2013), suggesting a broad role of this soluble ubiquitin ligase in ERAD.

Next, we tested the involvement of Cdc48 in the degradation of LD proteins. Inactivation of Cdc48 in cells expressing the temperature-sensitive allele *cdc48-6* (Schuberth & Buchberger, 2005) strongly delayed the degradation of Pgc1 (Fig 1C) and Dga1 (Fig EV3A). Moreover, *cdc48-6* cells accumulated large amounts of membrane-bound, ubiquitinated Pgc1 (Fig 1D). These findings are consistent with the well-characterized role of Cdc48 in releasing ubiquitinated ERAD substrates from the ER membrane into the cytoplasm for degradation by the proteasome (Bays *et al*, 2001; Ye *et al*, 2001; Braun *et al*, 2002; Jarosch *et al*, 2002; Rabinovich *et al*, 2002; Stein *et al*, 2014). However, it should be noted that a less stringent Cdc48 allele, *cdc48-3* (Latterich *et al*, 1995), while stabilizing other ERAD substrates including the Doa10 substrate Erg1, had a negligible effect on the degradation of Pgc1 and Dga1 (Fig EV3B and C). To resolve the inconsistency between the two *CDC48* mutant alleles on the degradation of Pgc1, we sequenced them. Interestingly, we found that *cdc48-3* is mutated in the D1 ATPase domain (P257L and R387K), whereas, in agreement with its tighter phenotype, *cdc48-6* contains mutations in both D1 (P257L) and D2 (A540T) ATPase domains. Thus, while membrane extraction of ER luminal and polytopic proteins requires wt Cdc48, ERAD of LD proteins needs only residual Cdc48 activity. This decreased requirement for Cdc48 suggests that membrane extraction of hairpin-containing proteins like Pgc1 and Dga1 may need only a single Cdc48 ATPase cycle while extraction of polytopic ERAD substrates might require processive and/or multiple rounds of Cdc48 activity.

Mutants with impaired proteasomal function, such as *pre2*, showed delayed elimination of Pgc1 and Dga1, indicating that their degradation is proteasome-dependent (Fig EV4; Heinemeyer *et al*, 1993). Consistent with the proteasome involvement, impairment of vacuolar proteolytic activity by *PEP4* mutation did not affect Pgc1 or Dga1 turnover (Figs EV1A and EV2C; Ammerer *et al*, 1986). Altogether, these data show that Pgc1, Dga1, and Yeh1 are *bona fide* ERAD substrates of the Doa10 complex.

### Pgc1 is a membrane-anchored protein stably associated with LDs

Pgc1 is a protein of unknown function predicted to have a glycerophosphodiester phosphodiesterase motif and to associate with membranes through a C-terminal hydrophobic segment (Beilharz *et al*, 2003; Fernandez-Murray & McMaster, 2005; Fisher *et al*, 2005). In recent proteomic analysis, it was identified as a high-confidence LD protein (Grillitsch *et al*, 2011; Currie *et al*, 2014). Indeed, when expressed from the constitutive alcohol dehydrogenase (*ADH1*) promoter as a N-terminal GFP fusion, GFP-Pgc1 localized to LDs in both wt and *doa10Δ* cells (Fig 2A). In the *doa10Δ* mutant, GFP-Pgc1 was also detected at the nuclear and cortical ER (Fig 2A). In most wt cells, GFP-Pgc1 was virtually undetectable at the ER. Next, we analyzed the role of the C-terminal hydrophobic region in Pgc1 membrane association (Fig 2B). Upon subcellular fractionation, endogenously expressed Pgc1 bearing a N-terminal HA epitope was found in the microsomal fraction (Fig 2C, mock), which under the isolation conditions also contains LDs. Importantly, the majority of Pgc1 maintained its microsomal

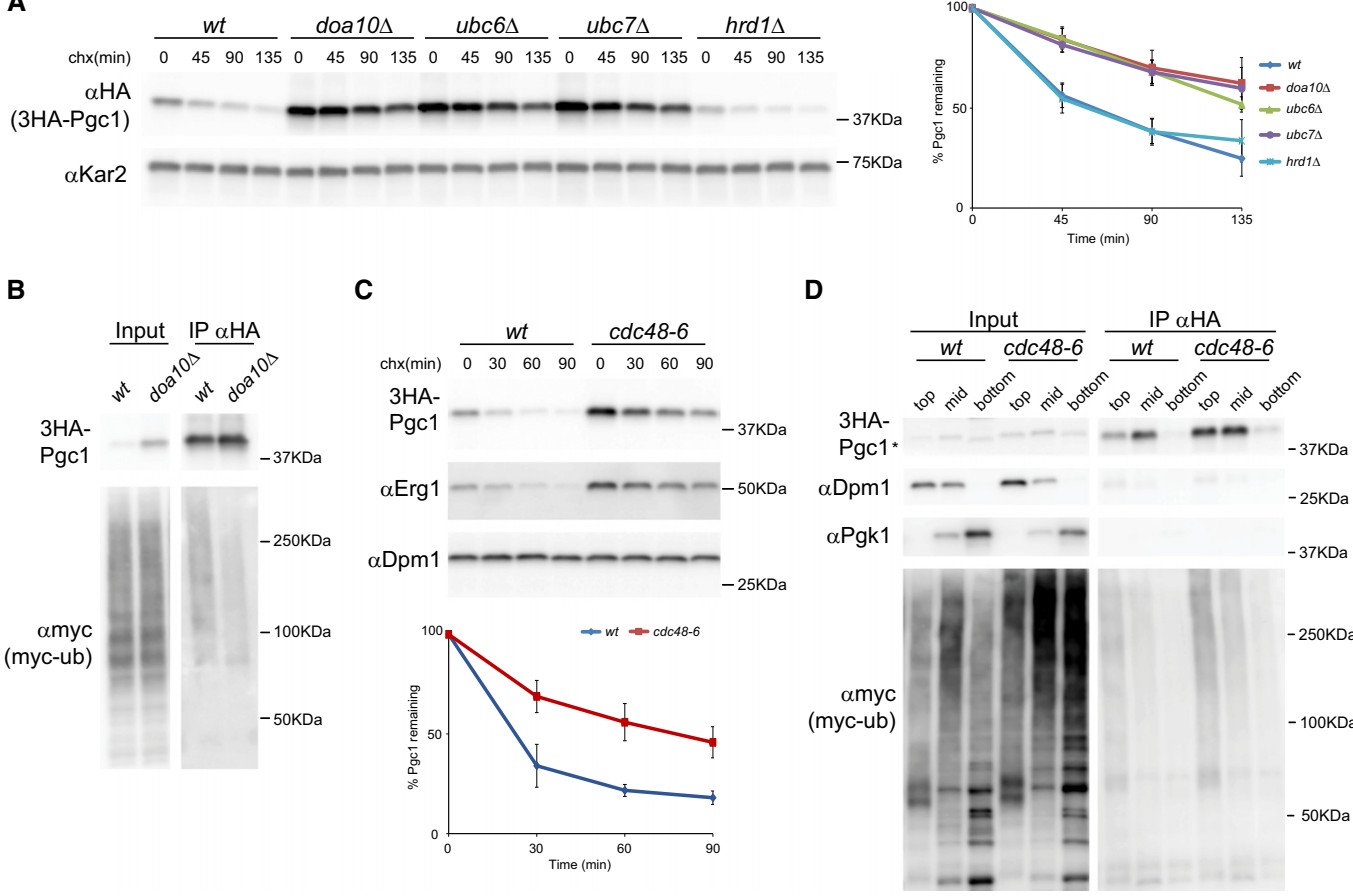

**Figure 1.  The LD protein Pgc1 is an ERAD substrate.**

A    The degradation of 3HA-Pgc1 was analyzed in cells with the indicated genotype upon inhibition of protein synthesis with cycloheximide (chx). A plasmid-borne 3HA-Pgc1 expressed from the endogenous promoter was used. 3HA-Pgc1 was detected with anti-HA antibodies. Kar2 was used as a loading control and detected with anti-Kar2 antibodies. The graph on the right shows the average of four independent experiments; error bars represent the standard deviation.

B    The ubiquitination of 3HA-Pgc1 was analyzed in cells with the indicated genotype and expressing myc-tagged ubiquitin. 3HA-Pgc1 was immunoprecipitated using anti-HA and polyubiquitin-conjugated 3HA-Pgc1 was detected using anti-myc antibody. Input fraction corresponds to 2% of the total amount used for IP.

C    The degradation of 3HA-Pgc1 was analyzed as in (A) in cells bearing the *CDC48* temperature-sensitive allele *cdc48-6*. Cells were grown 2.5 h at 37°C prior to addition of cycloheximide. Inactivation of Cdc48 mutant protein was confirmed by stabilization of the ERAD substrate Erg1 in the same cells. The graph shows the average of four independent experiments; error bars represent the standard deviation.

D    Membrane association of ubiquitinated 3HA-Pgc1 was analyzed by density gradient centrifugation followed by immunoprecipitation. Cells of the indicated genotype and expressing myc-tagged ubiquitin were grown as in (C). Lysates were subjected to centrifugation on an Optiprep density gradient. Collected fractions were analyzed by Western blotting before and after immunoprecipitation with anti-HA antibodies. Dpm1 and Pgk1 were used as markers for membrane-bound and soluble fractions, respectively. Input fraction corresponds to 10% of the total amount used for IP. "Top", "mid", and "bottom" indicate the fractions from the gradient. Asterisk marks a faster migrating unspecific band in the input bottom fraction.

Source data are available online for this figure.

association after alkaline treatment, which removes peripherally associated proteins such as Kar2, and it was only released upon detergent solubilization of membranes. A similar behavior was displayed by a truncated version encoding the last 47 amino acids of Pgc1 (3HA-GFP-Pgc1$^{275-321}$) encompassing the predicted hydrophobic region (Fig 2D). Thus, Pgc1 is anchored to membranes through its hydrophobic C-terminal region. *In silico* analysis through the server TOPCONS predicted that this hydrophobic C-terminus adopts a hairpin configuration, which is typical of many other LD proteins (including Dga1; Tsirigos *et al*, 2015). A number of hairpin-containing LD proteins have been shown to target the ER membrane before concentrating on LDs (Jacquier *et al*, 2011; Wilfling *et al*,

2013). Several lines of evidence indicate that Pgc1 displays a similar behavior. First, in cells lacking LDs, such as the *are1Δ are2Δ lro1Δ dga1Δ* mutant deficient in neutral lipid synthesis (Sandager *et al*, 2002; Sorger *et al*, 2004), Pgc1 co-localized with the ER marker Sec63-Cherry (Fig 3A). Second, experiments where LD formation was initiated upon galactose induction of *DGA1* expression in *are1Δ are2Δ lro1Δ* mutant (Jacquier *et al*, 2011) showed that ER labeling of GFP-Pgc1 decreases and the protein gradually concentrated on LDs as they formed. The kinetics of Pgc1 accumulation in LDs was comparable to the one of Erg6, a well-characterized LD marker protein (Fig 3B) (Grillitsch *et al*, 2011; Jacquier *et al*, 2011; Currie *et al*, 2014). To exclude that LD-associated Pgc1 had been recruited

from a cytosolic pool, similar LD induction experiments were performed in cells expressing Pgc1 fused to photoconvertible EOS (EOS-Pgc1) (McKinney *et al*, 2009). EOS-Pgc1 photoconverted at the ER was detected at LDs as these formed (Fig 3C). These experiments demonstrate that Pgc1 traffics through the ER *en route* to LDs. Importantly, the pool of Pgc1 at LDs is relatively stable, not diffusing back to the ER, as revealed in photobleaching experiments (Fig 3D), suggesting that the LD monolayer provides a more favorable environment to Pgc1 hydrophobic hairpin.

## Pgc1 is degraded by Doa10 at the ER

Since the Doa10 complex localizes exclusively at the ER, it would be expected that restricting Pgc1 to this organelle, as in the *are1Δ are2Δ lro1Δ dga1Δ* mutant, would result in its faster degradation. Indeed, Pgc1 degradation was significantly accelerated in this mutant (Fig 4A). In the absence of LDs, Pgc1 degradation was still

dependent on Doa10, as the protein was stabilized by additional mutation of this ubiquitin ligase while deletion of *HRD1* had no effect. On the other hand, expansion of LD surface by oleate feeding, decreased TAG lipolysis (in *tgl3Δ tgl4Δ tgl5Δ* cells) or both delayed Pgc1 turnover (Fig 4B). Importantly, the kinetics of degradation of Vma12-Ndc10C', a Doa10 substrate that does not localize to LDs (Furth *et al*, 2011), was unaffected indicating that the treatment affects specifically LD-localized Pgc1. These experiments show that Doa10 promotes the degradation of the ER pool of Pgc1 while LD-localized Pgc1 is spared from degradation.

## Pgc1 hydrophobic hairpin is necessary and sufficient for Doa10-dependent degradation

Next, we analyzed whether Pgc1 degradation required its hydrophobic hairpin. Derivatives of Pgc1 in which the hydrophobic hairpin (residues 275–321) was replaced by the membrane anchor (MA) of

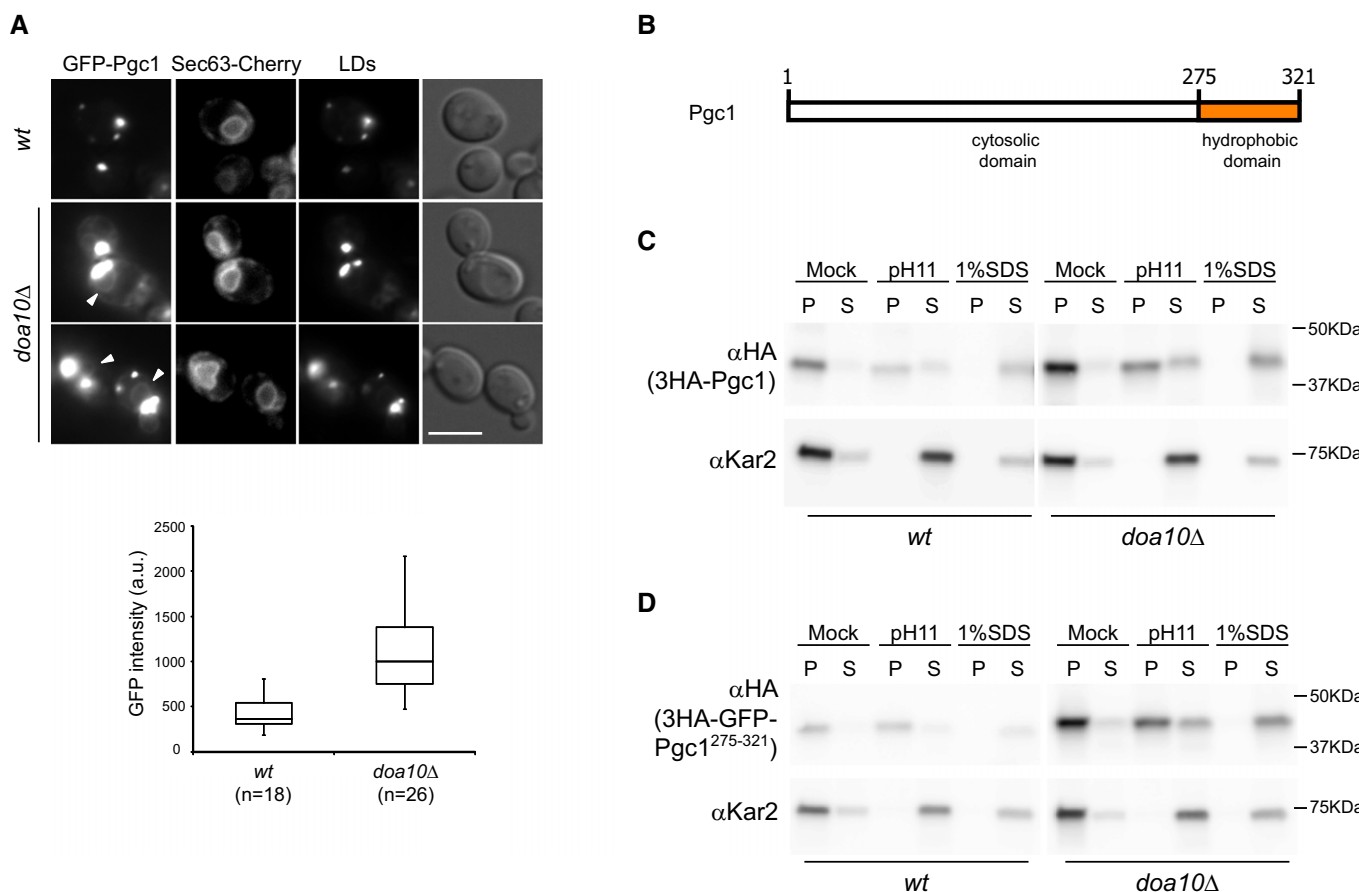

**Figure 2. Pgc1 behaves as an integral membrane protein.**

A  Localization of chromosomally expressed GFP-Pgc1 from the constitutive *ADH1* promoter in wt and *doa10Δ* cells. Arrowheads indicate GFP-Pgc1 labeling at the ER, which is stained by Sec63-Cherry. LDs were visualized upon staining with the neutral lipid dye MDH. On the bottom, box plot with quantitation of GFP-Pgc1 fluorescence intensity at the nuclear envelope in wt and *doa10Δ*. Horizontal lines indicate median, first and third quartiles of the distribution; whiskers indicate maximum and minimum values. GFP-Pgc1 measurements were taken as described in the Materials and Methods section. Scale bar: 5 μm.

B  Schematic representation of Pgc1. The location of the predicted hydrophobic domain is indicated.

C  Crude membranes from wt and *doa10Δ* cells expressing 3HA-Pgc1 were subjected to the indicated treatments and subsequently fractionated into membrane pellet (P) and supernatant (S).

D  Crude membranes from wt and *doa10Δ* cells expressing 3HA-GFP-Pgc1[275–321] were analyzed as in (C).

Source data are available online for this figure.

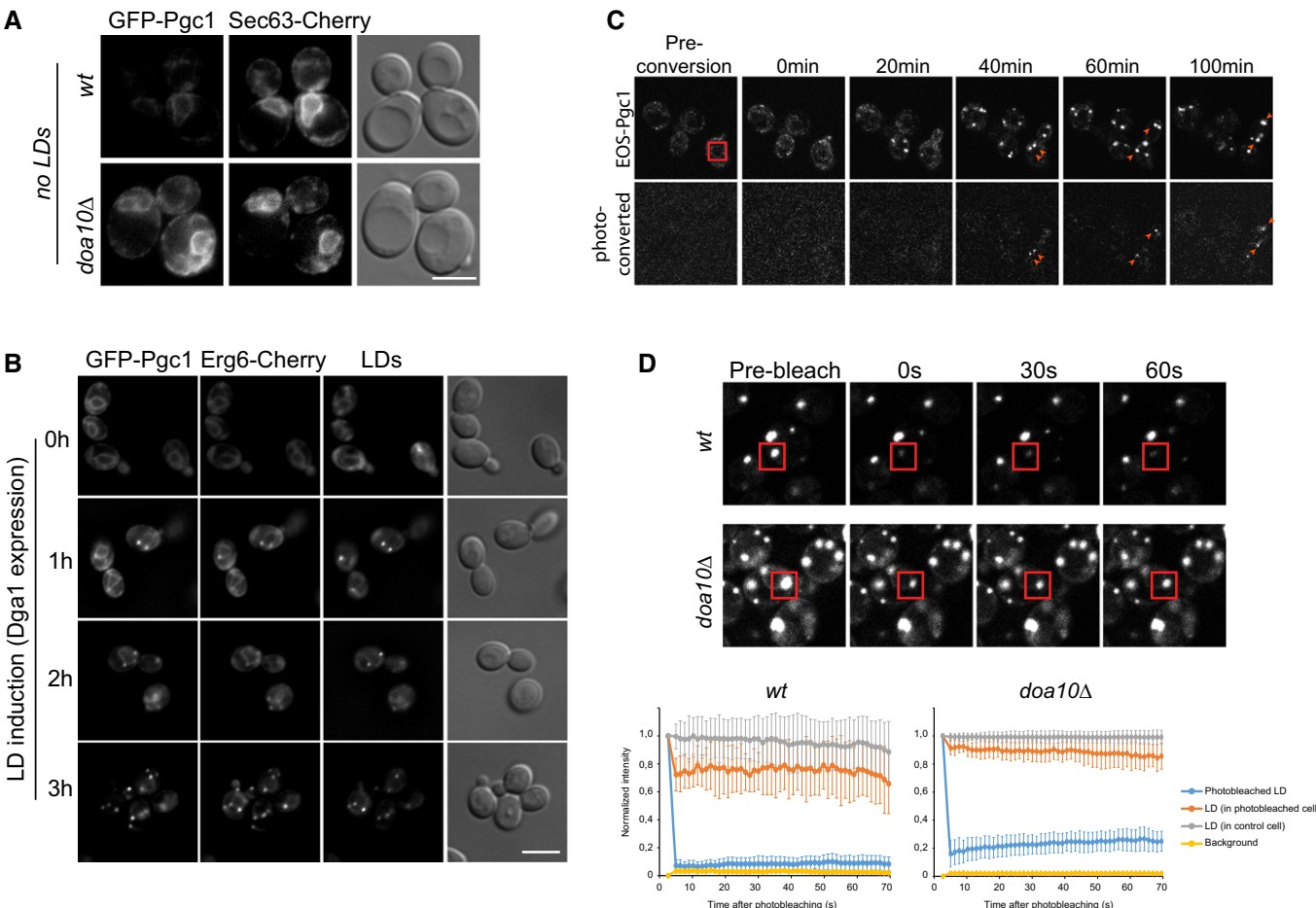

**Figure 3. Pgc1 localizes to the ER before stably concentrating on LDs.**

A   Localization of GFP-Pgc1 in *are1Δ are2Δ lro1Δ dga1Δ* cells in the presence (no LDs) or absence of *DOA10* (no LDs *doa10Δ*). The ER was visualized by expression of Sec63-Cherry. Scale bar: 5 μm.

B   GFP-Pgc1 was expressed under the *ADH1* promoter; LD formation was stimulated by galactose-induced expression of *DGA1* in *are1Δ are2Δ lro1Δ*. Fluorescence microscopy was used to follow GFP-Pgc1 localization over time. LDs were visualized upon staining with the neutral lipid dye MDH. Scale bar: 5 μm.

C   Photoconversion of tdEOS-Pgc1 expressed under the *ADH1* promoter; LD formation was stimulated by galactose-induced expression of *DGA1* in *are1Δ are2Δ lro1Δ doa10Δ* cells. The red square marks the photoconverted region. The time (in minutes) after photoconversion is indicated. Arrowheads point to LDs containing photoconverted tdEOS-Pgc1. Scale bar: 5 μm.

D   FRAP experiment of GFP-Pgc1 in wt and *doa10Δ* cells. Representative examples are shown. The bleached areas are marked by red squares and include a LD adjacent to ER. The time (in seconds) after photobleaching is indicated. Each graph shows average fluorescence intensities for 10 cells normalized to pre-bleached plotted over time. Error bars indicate standard deviation.

the ER proteins Scs2 (Pgc1[Scs2MA]; Loewen *et al*, 2007) or Bos1 (Pgc1[Bos1MA]; Lian & Ferro-Novick, 1993) were generated and their localization analyzed by fluorescence microscopy. In both wt and *doa10Δ* cells, the two chimeric proteins co-localized with the ER marker Sec63 and were excluded from LDs (Fig 5A). These data indicate that LD targeting of Pgc1 requires its MA. Despite their ER localization, the chimeric constructs Pgc1[Scs2MA] and Pgc1[Bos1MA] were stable, showing that Pgc1 hydrophobic hairpin is also required for the Doa10-mediated ERAD (Fig 5B). Conversely, the construct 3HA-GFP-Pgc1[275–321] encoding for Pgc1 hydrophobic hairpin was extremely short-lived in wt cells while its turnover was strongly delayed in *doa10Δ* mutants (Fig 5C). Thus, Pgc1 hydrophobic hairpin is necessary and sufficient for its Doa10-dependent degradation. The extremely short half-life of 3HA-GFP-Pgc1[275–321] in wt cells precluded its detection by fluorescence microscopy. In contrast, in

*doa10Δ* cells, 3HA-GFP-Pgc1[275–321] was readily detected and showed a dual localization, with a pool at the ER and another at LDs (Fig 5D). Altogether, these data indicate that Pgc1 hydrophobic hairpin is necessary and sufficient for its LD targeting. Moreover, they show that the same region of Pgc1 acts as degradation signal (or degron) for Doa10-mediated ERAD. The overlap of signals promoting LD localization and ERAD targeting offers the potential for regulating these competing events, for example, depending on the metabolic status of the cells.

### Hairpins of LD proteins can serve as degrons for Doa10 ERAD

Several proteins exchanging between ER and LDs were shown to associate with membranes through a hydrophobic hairpin (Jacquier *et al*, 2011; Wilfling *et al*, 2013). Among these are the

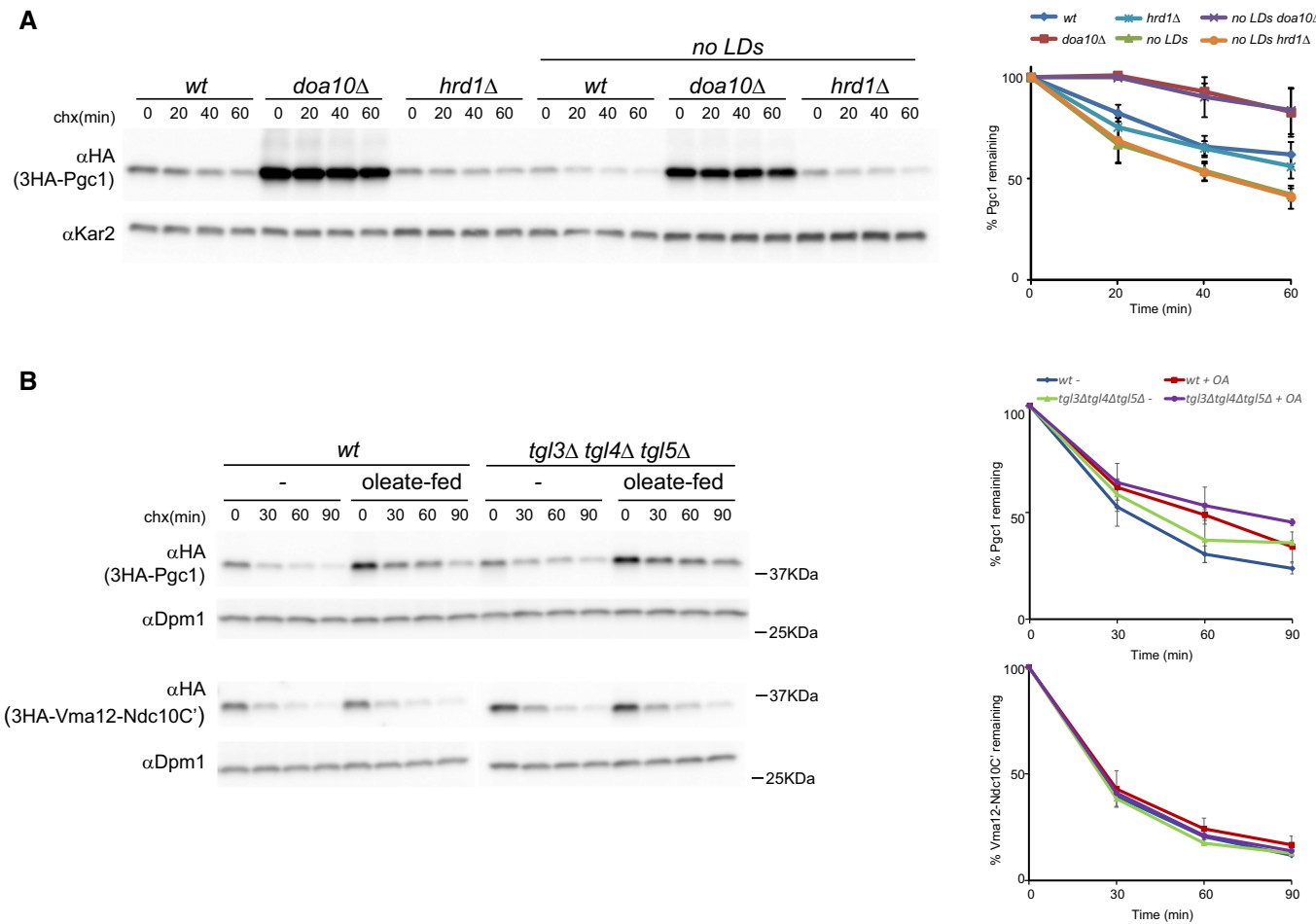

**Figure 4. Pgc1 is degraded by Doa10 at the ER.**

A  The degradation of 3HA-Pgc1 was analyzed in cells with the indicated genotype as in Fig 1A. The graph shows the average of three independent experiments; error bars represent the standard deviation.

B  The degradation of the Doa10 substrates 3HA-Pgc1 and 3HA-Vma12-Ndc10C' was analyzed in cells of the indicated genotypes treated with oleic acid. 3HA-Pgc1 and 3HA-Vma12-Ndc10C' were detected with anti-HA antibodies. Dpm1 was used as a loading control and detected with anti-Dpm1 antibodies. The graph shows the average of three independent experiments; error bars represent the standard deviation.

Source data are available online for this figure.

yeast lipases Tgl1 and Yeh1, the diacylglycerol acyltransferase Dga1, its mammalian homologue DGAT2, or the *Drosophila* GPAT4. Our results indicate that Pgc1 might be another of such proteins. Since Pgc1, Yeh1, and Dga1 are Doa10 substrates, we wondered whether hairpins mediating LD targeting of certain proteins could function as a general degradation signal for Doa10. To directly test this possibility, we replaced the hydrophobic hairpin of Pgc1 with the heterologous hairpin of GPAT4. This domain (residues 160–216) was shown to be sufficient to target mCherry to LDs in *Drosophila* cultured cells (Wilfling *et al*, 2013). While in wt cells 3HA-Pgc1-GPAT4$^{160–216}$ was a short-lived protein, deletion of *DOA10* strongly increased the half-life of the chimeric protein (Fig 6A). Like Pgc1, 3HA-Pgc1-GPAT4$^{160–216}$ was further stabilized in *doa10Δ ubr1Δ*. In contrast, deletion of *HRD1* had no effect on 3HA-Pgc1-GPAT4$^{160–216}$ degradation (Fig EV5). Importantly, the chimeric construct GFP-Pgc1-GPAT4$^{160–216}$ localized to LDs in wt cells, indicating that

GPAT4 hydrophobic hairpin is a functional LD targeting signal in yeast (Fig 6B). Moreover, conditions that strongly stabilized GFP-Pgc1-GPAT4$^{160–216}$, such as *doa10Δ ubr1Δ* mutant, lead to its accumulation at the ER besides LDs (Fig 6B). To further characterize the behavior of 3HA-Pgc1-GPAT4$^{160–216}$, we analyzed its degradation in the absence of LDs. Like Pgc1, the degradation of the chimeric protein was strongly accelerated in *are1Δ are2Δ lro1Δ dga1Δ* cells (no LDs) (half-life < 20′ in no LDs vs. ~45′ in wt cells; Fig 6C). Importantly, degradation of 3HA-Pgc1-GPAT4$^{160–216}$ in this background was substantially delayed by additional deletion of *DOA10* (Fig 6C). The longer half-life of Pgc1-GPAT4$^{160–216}$ in the mutant *are1Δ are2Δ lro1Δ dga1Δ doa10Δ* allowed us to confirm that in the absence of LDs, the GFP-tagged chimeric protein indeed localized to the ER, like full-length Pgc1 (Fig 6D). Thus, the chimera containing the well-characterized GPAT4 hairpin behaves as wt Pgc1, indicating that this LD targeting motif serves as a generic Doa10 degron.

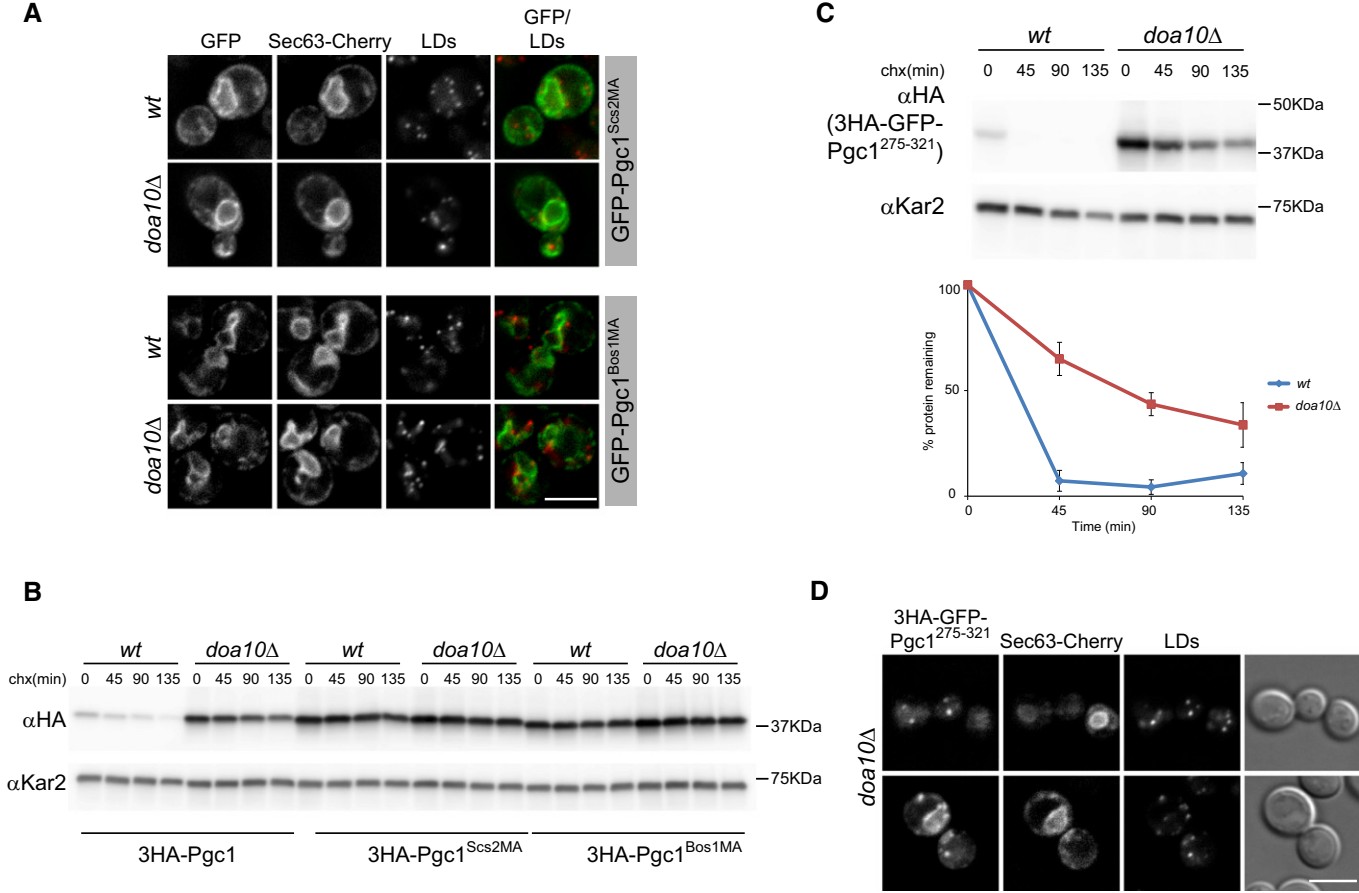

**Figure 5. Pgc1 hydrophobic hairpin is necessary and sufficient for Doa10-dependent degradation.**

A   Localization of GFP-tagged Pgc1 derivatives in which the hydrophobic hairpin (aa275–321) was replaced by membrane anchor of Scs2 (Pgc1$^{Scs2MA}$) or Bos1
    (Pgc1$^{Bos1MA}$). ER and LDs were visualized by expression of Sec63-Cherry and MDH staining, respectively. Scale bar: 5 μm.

B   The degradation of 3HA-Pgc1 or the indicated chimeras in wt and *doa10Δ* cells was analyzed as in Fig 1A. The blot is representative of at least three independent
    experiments.

C   The degradation of 3HA-GFP-Pgc1$^{275–321}$ in wt and *doa10Δ* cells was analyzed as in Fig 1A. The graph shows the average of four independent experiments; error bars
    represent the standard deviation.

D   Localization of 3HA-GFP-Pgc1$^{275–321}$ in *doa10Δ* cells. The ER was visualized by expression of Sec63-Cherry. Scale bar: 5 μm.

Source data are available online for this figure.

## Discussion

Here, we uncover a new class of substrates of the ERAD ubiquitin ligase Doa10. These are proteins that contain a hydrophobic hairpin and that localize to ER and LD membranes. By degrading specifically the ER pool, ERAD restricts their localization to LDs, thereby contributing to maintain the individual membrane identities of the ER and LDs. These findings reveal a function for the ERAD pathway that is distinct from its role in protein quality control or in lipid-dependent degradation of sterol enzymes (Ruggiano *et al*, 2014). We call this novel function "protein spatial control" since it leads to the degradation of a protein based on its localization rather than its folding status.

The involvement of quality control systems in the degradation of mislocalized proteins has been described in different contexts. For example, tail- and GPI-anchored proteins failing to insert in the ER membrane are selectively targeted for degradation (Hessa *et al*,

2011; Ast *et al*, 2014; Rodrigo-Brenni *et al*, 2014). Similarly, ER and peroxisomal proteins erroneously inserted in mitochondria outer membrane are degraded by an ill-defined process involving the ATPase Msp1/Atad1 (Chen *et al*, 2014; Okreglak & Walter, 2014). Thus, sequential, non-redundant quality control processes prevent proteins to accumulate in the inappropriate cellular compartment. The process described here expands this concept to proteins in continuous but distinct membrane regions, such as ER and LDs. Spatial control of LD proteins resembles the degradation of certain proteins in the INM by the Asi complex, a recently identified ERAD branch (Foresti *et al*, 2014; Khmelinskii *et al*, 2014). In this case, Asi-mediated ERAD excludes mislocalized proteins from the INM, therefore contributing to maintain the identity of this ER domain. We speculate that protein spatial control by ERAD might be a general mechanism to generate heterogeneity and/or functional domains, such as INM and LDs, in the continuous membrane of the ER.

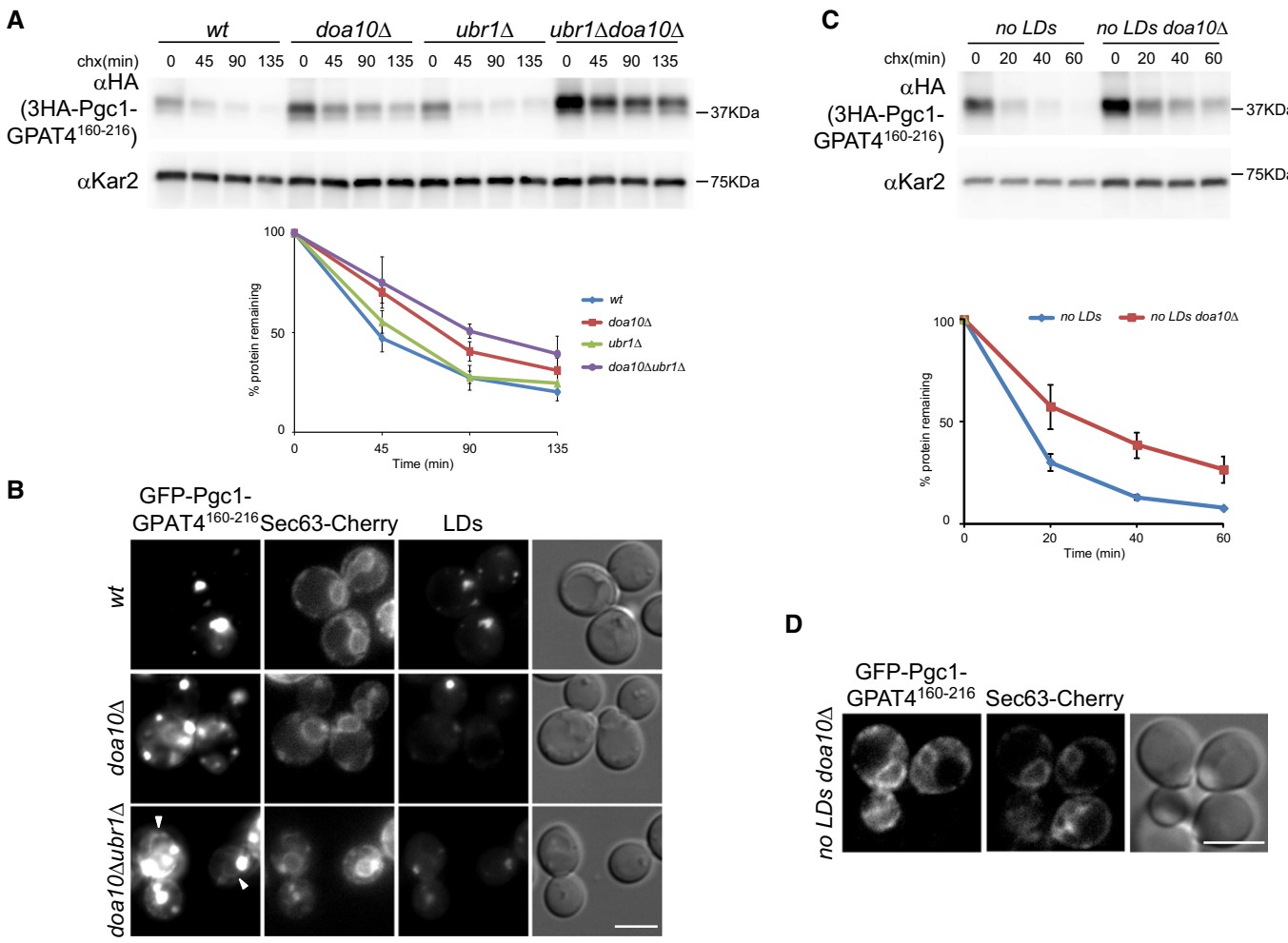

**Figure 6.  A heterologous hydrophobic hairpin functions as a Doa10 degron.**

A    The degradation of 3HA-Pgc1-GPAT4^160–216 containing GPAT4 hydrophobic hairpin (aa160–216) replacing the one of Pgc1 was analyzed in cells with the indicated genotype as in Fig 1A. The graph shows the average of three independent experiments; error bars represent the standard deviation.

B    Localization of GFP-Pgc1-GPAT4^160–216 in cells with the indicated genotype. Arrowheads indicate GFP-Pgc1-GPAT4^160–216 labeling at the ER, marked with Sec63-Cherry. LDs were visualized upon staining with MDH. Scale bar: 5 μm.

C    The degradation of 3HA-Pgc1-GPAT^160–216 in cells with the indicated genotype was analyzed as in Fig 1A. The graph shows the average of three independent experiments; error bars represent the standard deviation.

D    Localization of GFP-Pgc1-GPAT^160–216 in are1Δ are2Δ lro1Δ dga1Δ doa10Δ mutant (no LDs doa10Δ). The ER was visualized by expression of Sec63-Cherry. Scale bar: 5 μm.

Source data are available online for this figure.

How ERAD substrates are recognized based on their localization is not clear yet. In the case of LD proteins, the simplest possibility is that the hydrophobic hairpin, while in a bilayer membrane, may display some conformation instability or features typical of a misfolded protein, which target them to ERAD as "quality control" substrates. At the LD monolayer, the hairpins might adopt a more favorable conformation, decreasing the mobility of proteins back into the ER and as such keeping them away from the ERAD machinery and sparing them from degradation. Indeed, photobleaching experiments indicate that Pgc1 has a long dwell time at the LD monolayer. Similarly, GPAT4 is quite stable once at the LD monolayer (Wilfling *et al*, 2013). Given the overlap of the signals for LD targeting and Doa10 degradation, there is the potential for modulation of the two distinct outcomes by other factors. Such a mechanism might explain the changes of Dga1 localization under different metabolic states (Sorger & Daum, 2002; Markgraf *et al*, 2014). However, the identification of factors involved in the recognition of the hydrophobic hairpins is necessary to understand the process.

The regulation of LD proteins by ERAD is likely a conserved process in eukaryotes. A recent report showed that DGAT2 is also degraded by ERAD (Choi *et al*, 2014). Although in this case the ubiquitin ligase involved was Gp78, the homologue of the yeast Hrd1, the general process has remarkable similarities with our findings. The degradation of DGAT2 required its hydrophobic hairpin while a block in DGAT2 degradation led to its accumulation in the ER and increased TAG levels. Similarly, yeast cells lacking *DOA10*

display higher levels of TAG, in agreement with increased levels of Dga1 in this mutant (Fei *et al*, 2009; our unpublished data). Other hairpin-containing LD proteins are also involved in synthesis, modification, and turnover of lipids at the core and monolayer of LDs. Therefore, deregulation of their levels and distribution might account for the LD defects in *doa10Δ* cells. The ubiquitin ligase Ubr1 can to some extent compensate for the loss of *DOA10*, both in the degradation of LD proteins (our data) and of some ER membrane proteins (Stolz *et al*, 2013). These findings point to a more prominent role of a cytosolic ubiquitin ligase in ERAD, and future studies should address how Ubr1 is recruited to its ER membrane-bound substrates.

Additional links between ERAD and LDs have been previously reported, in particular the dynamic localization of some ERAD components to ER and LD membranes (Klemm *et al*, 2011; Spandl *et al*, 2011; Wang & Lee, 2012; Jo *et al*, 2013; Olzmann *et al*, 2013). If in most cases this has not been functionally dissected, for UBXD8, an adaptor for the Cdc48/p97 ATPase, its LD localization is important to regulate cellular TAG levels. Under conditions disfavoring lipolysis, UBXD8 relocates from the ER to LDs where it inhibits the activity of a major TAG lipase ATGL (Olzmann *et al*, 2013). Interestingly, this function of UBXD8 did not appear to involve proteolysis suggesting multiple and complex connections between ERAD and LD regulation that should be the focus of future studies.

# Materials and Methods

## Reagents

Rat monoclonal anti-hemagglutinin (HA) antibody (clone 3F10) was purchased from Roche and used at 1:2,000 dilution for immunoblot and 1:1,000 for immunoprecipitation; mouse anti-myc antibody was purchased from Roche and used at 1:1,000 dilution; rabbit polyclonal anti-GFP and anti-Kar2 antibodies were purchased from Santa Cruz Biotechnology and used at 1:1,000 dilution; anti-Pgk1 antibody was purchased from Invitrogen and used at 1:10,000 dilution; rat monoclonal (3H9) anti-GFP antibody was purchased from Chromotek and used at 1:2,000 dilution; anti-Dpm1 antibody was purchased from Life Technologies and used at 1:3,000 dilution; rabbit polyclonal anti-Erg1 antibody was raised against the full-length protein as described in Foresti *et al* (2013). Cycloheximide (Sigma-Aldrich) was used at 250 μg/ml. Monodansyl pentane (MDH) was purchased from Abgent and used at 0.1 mM. All other reagents and chemicals were purchased from Sigma-Aldrich.

## Yeast strains and growth

Yeast strains were isogenic to wild-type BY4741 (*Mata ura3Δ0 his3Δ1 leu2Δ0 met15Δ0*), BY4742 (*Matα ura3Δ0 his3Δ1 leu2Δ0 lys2Δ0*), FY251 (*Mata ura3-52 his3Δ200 leu2Δ1 trp1Δ63*), or DF5 (*Mata his3Δ200 leu2-3, 2-112 lys2-801 ura3-52 trp1-1*). Single or multiple deletion mutants were obtained by transformation using PCR-based homologous recombination (Longtine *et al*, 1998) or by crossing haploid cells of opposite mating types. The list of strains is available in Table EV1. Cells were grown in minimal medium supplemented with the appropriate amino acids. For galactose induction, cells were pre-cultured in 2% raffinose medium for 24 h and induced by 2% galactose in early logarithmic phase.

For the oleic acid treatment, logarithmic YPD cultures were grown in the presence of 0.1% oleic acid and 1% Brij-58 or 1% Brij-58 alone for at least 10 generations before cycloheximide addition.

## Plasmids

A complete list of the plasmids used in this study is available in Table EV2. To generate pPC882, *SEC63-mCHERRY* sequence was amplified from yPC4314 using primers 185-721 and cloned into pRS416 between XhoI and XbaI sites.

To generate pPC1040, *PGC1* promoter (550 bp) was amplified from BY4741 genomic DNA with primers 1515-1516; *3HA-PGC1* was amplified with its own terminator from yPC6800 genomic DNA with primers 1517-1518. The fusion PCR product obtained with primers 1515-1518 (introducing SacI and PstI restriction sites, respectively) was cloned into pRS315 between SacI and PstI sites. To generate pPC1051, *PGC1* promoter was amplified from BY4741 genomic DNA with primers 1515-2091; *GFP-PGC1* was amplified with its own terminator from yPC6834 genomic DNA with primers 2092-1518. The fusion PCR product obtained with primers 1515-1518 was cloned into pRS315 between SacI and PstI sites.

To generate pPC1145, *ADH1* promoter-*GFP-PGC1* was amplified from yPC6834 genomic DNA with primers 1779-1518. The PCR product was cloned into pRS315 between SacI and PstI sites.

The DNA sequence encoding aa160–216 from GPAT4 was amplified from *Drosophila* S2 cells cDNA using primers 2066-2067. Other plasmids encoding *PGC1* are derived from pPC1040, pPC1051, and pPC1145 via sub-cloning, fusion PCR, or site-directed mutagenesis.

To generate pPC1196, *ADH1* promoter-*DGA1-GFP* was amplified from yPC7249 genomic DNA with primers 1779-185. PCR product was cloned into pRS415 between SacI and XhoI sites.

To generate pPC1299, *YEH1-3HA* was amplified with its own promoter from yPC9214 genomic DNA with primers 185-2148. The PCR product was cloned into pRS316 between XhoI and NotI sites.

The primers used are listed in Table EV3.

## Cycloheximide shut-off experiments

Cycloheximide shut-off experiments in exponentially growing cells ($OD_{600} \leq 1$) were performed at 30°C, unless differently specified. Whole-cell extracts for each time point were prepared as in Kushnirov (2000) and analyzed by SDS–PAGE and Western blot.

## Microsome preparation and alkaline extraction

Microsomes were prepared from exponentially growing cells ($OD_{600} = 1$) essentially as described in Liu *et al* (2011) and resuspended in 10 mM Hepes pH 7.4. For extraction of membrane proteins, equal amounts of microsomes were treated with 10 mM Hepes pH 7.4, or 0.2 M $Na_2CO_3$ pH 11 in water for 1 h at 4°C or 1% SDS in 10 mM Hepes for 1 h at room temperature. After incubation, samples were separated into pellet and supernatant by centrifugation at 100,000 *g*. Supernatant fractions were TCA-precipitated. Pellets were resuspended in Laemmli buffer and analyzed by SDS–PAGE and Western blot.

## Detection of ubiquitinated Pgc1

Logarithmically growing cells were harvested and washed with 10 mM sodium azide. Cell lysis was performed with glass beads in 0.8% SDS, 50 mM Tris–HCl pH 7.4 buffer containing 20 mM NEM and 0.1 mM PMSF. Pgc1 was immunoprecipitated with anti-HA antibodies in 0.2% SDS, 1% NP-40, 50 mM Tris–HCl pH 7.4. Immunoprecipitated material was analyzed by SDS–PAGE and Western blotting.

## Detection of membrane-associated ubiquitinated Pgc1

Membrane floatation was performed essentially as described (Bagnat *et al*, 2000). Fractions were collected and proteins precipitated with TCA. The pellets were resuspended in buffer containing 2 M urea, 50 mM Tris–HCl pH 7.4, 1 mM EDTA, 1% SDS and heated at 65°C for 10 min. After dilution with a buffer containing 50 mM Tris–HCl pH 7.4, 150 mM NaCl, 1 mM EDTA, 1% Triton X-100, Pgc1 was immunoprecipitated using anti-HA antibodies. Immunoprecipitated material was analyzed by SDS–PAGE and Western blotting.

## Microscopy

Fluorescence microscopy was performed at room temperature in a Zeiss Cell Observer HS equipped with a CMOS camera (Hamamatsu ORCA-Flash 4.0) controlled by 3i Slidebook 6.0 software. A $100\times$ 1.40 oil immersion objective was used. GFP, mCherry, and MDH signals were detected using GFP, RFP, and DAPI filters, respectively, with standard settings.

Cells were imaged in logarithmic growth phase. Images were acquired using the same settings, and brightness and contrast were processed in a similar manner. For GFP-Pgc1-expressing cells, GFP pixel intensities at the NE were measured using the line tool in ImageJ on a single-plane image. Intensity values were adjusted for background and were used to calculate median, first and third quartiles of the distribution for a set of images (wt or *doa10Δ*). Data were displayed in a box plot; whiskers extend to the highest and lowest value of the distribution.

Photobleaching experiments were performed on cells grown in YPD medium. Early stationary cells were diluted into the same medium to $OD_{600}$ 0.2 and grown up to $OD_{600}$ 1.2 before being transferred to a concanavalin A-pre-treated chamber. Live imaging was performed on a confocal Leica TCS SP5 microscope using a HCX PL APO CS100 $\times$ 1.40 oil objective and controlled by the LAS AF software.

Bleaching experiments were performed using the point bleach option of the FRAP module. Photobleaching was applied to LD in contact with the nuclear envelope. Three pre-bleach images were acquired followed by 800 ms of photobleaching at 80% laser power. Images were acquired every 1.32 s.

For analysis, the fluorescence intensity of four regions of interest was measured: the photobleached LD, a non-photobleached LD in the bleached cell to check for diffusion-dependent changes in fluorescence, a LD in a cell that was not photobleached to check for overall fluorescence variation, and region outside of the irradiated cell to check for overall background fluorescence. The fluorescence recovery values of the bleached region were background-subtracted and normalized to the average of the pre-bleaching values. The normalized recovery values were plotted after adjusting for the slow decay of fluorescence caused by imaging using areas of the image distant from the bleached region as described (Shibata *et al*, 2008). For each genotype, average and standard deviation were calculated from 10 photobleaching events.

For photoconversion experiments, cells were imaged using a laser scanning confocal microscope Leica TCS SP5 AOBS (inverted) with a $63\times$/1.4 oil immersion objective, using 13% of argon laser intensity (488 nm line) and 10% of DPPSS 561 laser intensity (561 nm line) at 30% output. Photoconversion was applied on a ROI as indicated in the figure with the laser 405 diode (405 nm line) at 10% laser intensity during 10 s. After conversion, a single image was taken every 20 min. Images were analyzed using ImageJ.

**Expanded View** for this article is available online.

## Acknowledgements

We thank E. Sabidó and C. Chiva for help with the mass spectrometry, Y. Barral, B. Crosas, and S. Jentsch for yeast strains and plasmids, and O. Foresti for discussion and critical reading of the manuscript. A. Ruggiano was supported by a "La Caixa" graduate fellowship; P. Carvalho is supported by CRG, an International Early Career Award from the HHMI, the EMBO Young Investigator Program, and grants from the Spanish MCCIN and ERC (FP7/2007-2013 ERC grant agreement no. 309477 DROPFAT). We acknowledge support of the Spanish Ministry of Economy and Competitiveness, "Centro de Excelencia Severo Ochoa 2013–2017", SEV-2012-0208.

## Author contributions

PC conceived and supervised the project. PC, AR, and GM designed the experiments and analyzed the data. AR and GM performed most of the experiments. LB performed the FRAP and photoconversion experiments. All authors discussed the results. PC and AR wrote the manuscript with input from GM.

## Conflict of interest

The authors declare that they have no conflict of interest.

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
