## [Review Process File · The EMBO Journal]

Manuscript EMBO-2015-93106

Spatial control of lipid droplet proteins by the ERAD ubiquitin ligase Doa10

Annamaria Ruggiano, Gabriel Mora, Laura Buxó and Pedro Carvalho

Corresponding author: Pedro Carvalho, Center for Genomic Regulation

Review timeline:

Submission date:	21 September 2015
Editorial Decision:	29 October 2015
Revision received:	01 May 2016
Editorial Decision:	23 May 2016
Revision received:	01 June 2016
Accepted:	02 June 2016

Editor: Andrea Leibfried

Transaction Report:

1st Editorial Decision

29 October 2015

Thank you for submitting your manuscript to us. It has now been seen by three referees, whose comments are provided below.

All referees appreciate your analyses. However, they also raise several concerns that currently preclude publication here. Referee #1 points out that further insight is needed to explain why LD proteins are targeted for degradation when in the ER but not in the LDs. Referee #2 and #3 both think that the described Cdc48 independency needs further support and that alternative possibilities should be discussed.

Given the constructive comments provided by the referees, I would like to invite you to provide a revised version of your manuscript addressing all concerns raised, especially the ones I noted above. Please also note that we require graphical display of both data points obtained whenever $n=2$ (point 4 of referee #3; e.g. in your figure 1A). I should add that it is EMBO Journal policy to allow only a single round of revision, and acceptance of your manuscript will therefore depend on the completeness of your responses in this revised version.

Thank you for the opportunity to consider your work for publication. I look forward to your revision.

REFeree COMMENTS

Referee #1:

Ruggiano, Carvalho et al

EMBO J

This manuscript describes a body of evidence that the ERAD ubiquitin ligase Doa10 is important for consolidating the LD-specific localization of certain LD resident proteins. This question of LD-specific localization is an interesting one because the phospholipid shell that encases LDs is contiguous with the cytoplasmic leaflet of the ER and many LD proteins are inserted into the ER prior to being sorted into LDs. The basic findings of this MS are that Doa10 plays an important ubiquitin- and proteasome-dependent function in degrading LD proteins in the ER, that the recognition maps closely to the hydrophobic hairpin structures that promote sorting to LDs, is apparently independent of the folding state of the LD protein substrate, and independent of the CDC48 ATPase complex normally required for extraction of ER substrates from ER membranes. The authors synthesize the data into a concept of 'spatial control' as a determinant for how organelle identity is established or maintained in eukaryotic cells.

Overall, the work is solid and the experiments reported are of well designed. The work is basically well-done and ultimately deserving of publication. However, this referee considers the work to be too limited in scope for it to be appropriate for EMBO J. General comments follow:

(i) The 'spatial control' idea is not new. Versions of this model have been invoked in apical vs basolateral membrane protein sorting in epithelial cells, in endosomes, and in the inner membrane of the nuclear envelope (as pointed out by the authors).

(ii) As it stands, the MS extends considerable data in establishing the Doa10 substrates of interest (Dga1, Yeh1 and particularly Pgc1) as bona fide ERAD substrates using standard and well-established assays. The demonstrations that the degradation occurs in ER (in strains where LD biogenesis is blocked), while interesting and a nice experiment, by itself defines a rather incremental step forward.

(iii) The most interesting aspect to this reviewer, and one that would elevate impact of this MS if further developed, is what are the activities that recognize ER-localized LD proteins as targets for degradation and how are these substrates extracted from LD surfaces? Can the authors define what it is about the hydrophobic hairpin in ER membranes that make it a degron whereas it is not a degron for LD-localized proteins? Can ERAD recognize unfolded protein substrates if these are sorted into LDs by helical hairpins? The authors have a sophisticated understanding of the problem and raise some of these questions explicitly as the questions identify novel directions, but they do not address these issues at all. The work described sets up these questions nicely, but there is no follow-through and this is what it would take, in this Reviewer's opinion, to produce a more impactful story.

Referee #2:

This study has followed up on hits from an earlier SILAC screen for protein products whose degradation relies on known ERAD pathway factors. Several lipid-droplet (LD) proteins were found in the screen to be increased in Doa10 deletion strains, an observation validated here. The authors use one of these substrates, Pgc1, to develop a model in which it inserts into the ER en route to LDs, with any molecules remaining in or returning to the ER being degraded by a Doa10-dependent pathway. Thus, degradation of mislocalized proteins improves the spatial segregation of certain LD proteins. This degradation is mediated by the hydrophobic region, which is found to be necessary and sufficient. Other hydrophobic regions of similar hairpin topology also seem to be Doa10 clients. Of note, the degradation pathway found here is slightly unusual in not requiring Cdc48 complex, but being partially dependent on Ubr1.

Overall, the paper is technically excellent, the writing is very clear, and the conclusions should be of interest to the EMBO readership. I generally support its publication, but have reservations about two aspects of the authors' main conclusion that warrants clarification.

1) The authors' favored model is one where Pgc1 inserts into the ER, then traffics to LDs, the failure of which triggers its ERAD via Doa10. This is a reasonable explanation of the observations, but it seems to me an alternative explanation is also plausible. In this alternative view, Pgc1 is normally directly inserted into LDs from the cytosol. If this fails (e.g., when LDs are absent or simply due to some inefficiency), PGC1 is then degraded via cytosolic quality control. Note that cytosolic QC of proteins with hydrophobic regions can be mediated by Doa10 (see Metzger et al., 2008; PMID 18812321), which would explain why Doa10 is needed, and why one can see an ER localization pattern. I favor this model because it would also explain why Cdc48 is NOT needed, and why the cytosolic ligase Ubr1 also contributes to the degradation, observations that in the authors' model are a little puzzling. This alternate model should certainly be considered and discussed. If a straightforward experiment exists to discriminate the two views, it is worth including.

2) Related to point 1, the authors' experiment to demonstrate Pgc1 traffic from the ER to LDs in Fig. 3A would appear to have an alternative explanation. In short, Fig. 3A is not a pulse-chase type experiment and hence one cannot be convinced that the ER Pgc1 observed at time 0 moves to the LDs observed at time 3h. It is equally possible that the ER population was degraded over this time (completely consistent with degradation kinetics shown in Fig. 1) while at the same time newly synthesized Pgc1 populates the newly formed LDs. It should be possible to discriminate these possibilities. One way is if photo-activatable GFP is used: activate the ER population at time 0 and follow it during LD induction. There may be other ways to test this as well.

The above two points need to be clarified before publication. Note that either model is interesting and worth publishing. My proposed alternative model would perhaps be less novel as earlier work has described the phenomenon of mislocalized proteins begin degraded in a mammalian system (PMID 24981174 and 21743475), although the factors are different.

Referee #3:

The manuscript by Ruggiano et al. reports a new function for the endoplasmic reticulum associated degradation (ERAD) pathway. ERAD is a protein quality control mechanism that eliminates misfolded proteins from the endoplasmic reticulum to maintain protein homeostasis. In budding yeast, it employs two major ubiquitin ligases, Hrd1p and Doa10p. The author followed up on their previous proteomic study, which identified a few lipid droplet proteins as potential substrates of the ERAD ubiquitin ligase Doa10 because the expression of these proteins was elevated in Doa10 deletion yeast cells. In this study, the authors demonstrate that this is due to increased stability. They showed that the lipid droplet (LD) protein Pgc1, Dga1, Yeh1 are unstable proteins whose degradation depends on the 26S proteasome. The degradation of these proteins also requires Doa10 and the cognate ubiquitin conjugating enzyme Ubc6p and Ubc7p, but the Cdc48p ATPase is dispensable. To characterize this process, they focused on Pgc1. They showed that Pgc1 is normally partitioned between ER and lipid droplet and only the ER pool is subject to degradation by the ERAD pathway. They further show that the long transmembrane domain of Pgc1 is both necessary and sufficient for degradation by the proteasome. The study identifies a new class of substrate for the ERAD pathway (some lipid droplet proteins mislocalized to the ER membrane, and suggests that ERAD may have a quality control function to remove mislocalized LD proteins. Overall, the conclusions are well supported by the results. I only have a few relatively minor points for the authors to consider, which I believe will make the paper a stronger candidate for EMBO.

1. The authors tested the degradation of Pgc1 in Cdc48 and Npl4 mutant yeast strains and found no effect. They mentioned that the strains were characterized previously by testing a bona fide ERAD substrate, whose degradation is strongly inhibited by mutations in Cdc48 gene. However, that was done separately. It would be more convincing if the authors can put Erg1 together with Pgc1 in the same strain and show that one is affected by not the other. I am a little bit concerned that the negative data here might be due to some trivial experimental error because in a paper cited by the authors, a mammalian LD protein was recently demonstrated to be an ERAD substrate, but the degradation requires p97, the mammalian homolog of Cdc48.

2. In Figure 4C, the authors wish to demonstrate that oleate acid feeding, a procedure that induces LD formation may stabilize Pgc1. However, the difference is too small to be significant. It may be better to compare the stability of Pgc1 in the mutant yeast cells that do not have LD (those shown in Figure 3A before induction) and those that have the Dga1 expression induced (LD induction

condition shown in Figure 3A) or perform the oleate acid feeding experiment in mutant cells that do not have any LD to begin with.

Minor points:

1. On page 9, the authors mentioned Erg6 as "a well-characterized LD marker protein". When I searched PubMed, I could not find many papers showing this as an LD marker. Please give the reference.
2. On page 13, the authors mentioned that their study reveals "a function for the ERAD pathway that is distinct from its role in protein quality control". In a broad sense, the newly demonstrated ERAD function can still be considered as quality control as it removes LD proteins mislocalized to the ER, which is analogous to the cytosolic quality control pathway that deals with mislocalized ER membrane proteins.
3. Figure 1B, what do I and U stand for? Please explain.
4. Can standard deviation be calculated from two experimental datasets? It may be more appropriate to say that the error bars indicate the range of two experimental repeats. With key experiments (such as those shown in Figure 4B), it would be more convincing if the authors provides 3 repeats instead of 2, so can add p value to the graph. Some quantification graphs appear too busy. It may be better to separate the curves into 2 or more panels, so the comparison is more obvious? The FRPA experiments need to explain the number of LD analyzed in the figure legend.

1st Revision - authors' response

01 May 2016

Response to Reviewers

We were very pleased to see that the reviewers found our study interesting, technically excellent and were generally supportive of publication in the EMBO Journal.

We are now submitting a revised manuscript in which the issues raised have been addressed. In particular, this version includes two additional key results. First, we took advantage of live cell microscopy and photoconversion experiments to follow a population of Pgc1 molecules. The results demonstrate that Pgc1 traffics through the ER *en route* to LDs and argue against Pgc1 being independently targeted to ER and LDs. Second, we show that the Cdc48 ATPase function is necessary for Doa10-mediated degradation of Pgc1 and Dga1, which are strongly stabilized in cells expressing the tight allele *cdc48-6*. Importantly, we show that in *cdc48-6* cells, polyubiquitinated Pgc1 accumulates and partitions with ER membranes, as assayed by membrane flotation analysis. These results are consistent with the role of Cdc48 in Pgc1 membrane extraction. In the previous version of the manuscript, we showed that the kinetics of Pgc1 and Dga1 degradation was essentially indistinguishable between *wt* and cells expressing *cdc48-3* allele, even if the steady state of both substrates were higher in the mutant cells (this data is still presented as a supplemental figure). Our interpretation of these results is that residual Cdc48 activity in *cdc48-3* cells is sufficient for ER extraction of LD proteins that associate with the membrane through a hydrophobic hairpin. Perhaps membrane extraction of hairpin-containing proteins like Pgc1 and Dga1 may need only a single Cdc48 ATPase cycle while extraction of polytopic ERAD substrates might require processive and/or multiple rounds of Cdc48 activity. The ATPases of proteasome 19S regulatory particle have also been implicated in membrane extraction of some ERAD substrates. However we did not find any convincing evidence for a role of 19S ATPases in Pgc1 and Dga1 degradation. Finally, the number of replicates were increased for all the experiments. We believe that the new data, together with additional minor points listed below, make the paper significantly stronger.

Reviewer 1:

This manuscript describes a body of evidence that the ERAD ubiquitin ligase Doa10 is important for consolidating the LD-specific localization of certain LD resident proteins. This question of LD-specific localization is an interesting one because the phospholipid shell that encases LDs is

contiguous with the cytoplasmic leaflet of the ER and many LD proteins are inserted into the ER prior to being sorted into LDs. The basic findings of this MS are that Doa10 plays an important ubiquitin- and proteasome-dependent function in degrading LD proteins in the ER, that the recognition maps closely to the hydrophobic hairpin structures that promote sorting to LDs, is apparently independent of the folding state of the LD protein substrate, and independent of the CDC48 ATPase complex normally required for extraction of ER substrates from ER membranes. The authors synthesize the data into a concept of 'spatial control' as a determinant for how organelle identity is established or maintained in eukaryotic cells.

Overall, the work is solid and the experiments reported are of well designed. The work is basically well-done and ultimately deserving of publication. However, this referee considers the work to be too limited in scope for it to be appropriate for EMBO J. General comments follow:

(i) The 'spatial control' idea is not new. Versions of this model have been invoked in apical vs basolateral membrane protein sorting in epithelial cells, in endosomes, and in the inner membrane of the nuclear envelope (as pointed out by the authors).

(ii) As it stands, the MS extends considerable data in establishing the Doa10 substrates of interest (Dga1, Yeh1 and particularly Pgc1) as bona fide ERAD substrates using standard and well-established assays. The demonstrations that the degradation occurs in ER (in strains where LD biogenesis is blocked), while interesting and a nice experiment, by itself defines a rather incremental step forward.

(iii) The most interesting aspect to this reviewer, and one that would elevate impact of this MS if further developed, is what are the activities that recognize ER-localized LD proteins as targets for degradation and how are these substrates extracted from LD surfaces? Can the authors define what it is about the hydrophobic hairpin in ER membranes that make it a degron whereas it is not a degron for LD-localized proteins? Can ERAD recognize unfolded protein substrates if these are sorted into LDs by helical hairpins? The authors have a sophisticated understanding of the problem and raise some of these questions explicitly as the questions identify novel directions, but they do not address these issues at all. The work described sets up these questions nicely, but there is no follow-through and this is what it would take, in this Reviewer's opinion, to produce a more impactful story.

1- We agree with the reviewer that, in general, the concept of spatial segregation of proteins leading to membrane heterogeneity has been described in different contexts. However, to our knowledge, the INM and the LDs proteins (described here) are the only examples in which this segregation is facilitated by spatially restricted protein degradation. In both cases ERAD is involved suggesting that this pathway, besides its well-characterized function in degrading misfolded proteins, also plays important roles in determining ER architecture. Moreover our work expands the knowledge on the degradation of mislocalized proteins. Previous work showed that proteins that fail to target or that target the wrong membrane are detected as mislocalized (this work is now discussed and appropriate references included). We now show that cells also have the means to discriminate mislocalized proteins, even if these are in the same membrane, and raise the possibility that this spatial quality control is important in generating functional subdomains in a continuous membrane system, as is the case of the ER.

2 & 3- As pointed out by the reviewer, we convincingly demonstrate that only the ER pool of Pgc1 is targeted for Doa10-mediated degradation. We also show that Pgc1 hydrophobic hairpin is necessary and sufficient for the degradation and that the hairpin from a heterologous LD protein has a similar behaviour suggesting that these motifs might act as generic Doa10 degrons. Using FRAP and photoconversion experiments, we show that Pgc1 traffics through the ER before concentrating on LDs, from where it does not diffuse

efficiently back into the ER. On the other hand, the fact that Doa10 is a polytopic membrane protein (with multiple luminal loops) precludes its diffusion into LDs. All together these data lead to a simple model in which the LD pool of Pgc1 is physically separated from the ERAD machinery and as such protected from degradation. This model is simply based on the physical separation of the substrate from its degradation machinery. With the tools used in our study we cannot pinpoint the hairpin structural features. However, we postulate that hairpins that evolved to localize to LD monolayers (such as the ones of Pgc1 and Dga1) while in the ER bilayer reveal some conformational instability. As a consequence, they are recognized by Doa10 as quality control substrates. If and how ERAD recognizes misfolded substrates targeted to the LD surface by amphipathic helices, we agree with the reviewer that it is an interesting question. However, we believe it is outside of the scope of this study.

Reviewer 2:

This study has followed up on hits from an earlier SILAC screen for protein products whose degradation relies on known ERAD pathway factors. Several lipid-droplet (LD) proteins were found in the screen to be increased in Doa10 deletion strains, an observation validated here. The authors use one of these substrates, Pgc1, to develop a model in which it inserts into the ER en route to LDs, with any molecules remaining in or returning to the ER being degraded by a Doa10-dependent pathway. Thus, degradation of mislocalized proteins improves the spatial segregation of certain LD proteins. This degradation is mediated by the hydrophobic region, which is found to be necessary and sufficient. Other hydrophobic regions of similar hairpin topology also seem to be Doa10 clients. Of note, the degradation pathway found here is slightly unusual in not requiring Cdc48 complex, but being partially dependent on Ubr1.

Overall, the paper is technically excellent, the writing is very clear, and the conclusions should be of interest to the EMBO readership. I generally support its publication, but have reservations about two aspects of the authors' main conclusion that warrants clarification.

1) The authors' favored model is one where Pgc1 inserts into the ER, then traffics to LDs, the failure of which triggers its ERAD via Doa10. This is a reasonable explanation of the observations, but it seems to me an alternative explanation is also plausible. In this alternative view, Pgc1 is normally directly inserted into LDs from the cytosol. If this fails (e.g., when LDs are absent or simply due to some inefficiency), PGC1 is then degraded via cytosolic quality control. Note that cytosolic QC of proteins with hydrophobic regions can be mediated by Doa10 (see Metzger et al., 2008; PMID 18812321), which would explain why Doa10 is needed, and why one can see an ER localization pattern. I favor this model because it would also explain why Cdc48 is NOT needed, and why the cytosolic ligase Ubr1 also contributes to the degradation, observations that in the authors' model are a little puzzling. This alternate model should certainly be considered and discussed. If a straightforward experiment exists to discriminate the two views, it is worth including.

2) Related to point 1, the authors' experiment to demonstrate Pgc1 traffic from the ER to LDs in Fig. 3A would appear to have an alternative explanation. In short, Fig. 3A is not a pulse-chase type experiment and hence one cannot be convinced that the ER Pgc1 observed at time 0 moves to the LDs observed at time 3h. It is equally possible that the ER population was degraded over this time (completely consistent with degradation kinetics shown in Fig. 1) while at the same time newly synthesized Pgc1 populates the newly formed LDs. It should be possible to discriminate these possibilities. One way is if photo-activatable GFP is used: activate the ER population at time 0 and follow it during LD induction. There may be other ways to test this as well.

The above two points need to be clarified before publication. Note that either model is interesting and worth publishing. My proposed alternative model would perhaps be less novel as earlier work

has described the phenomenon of mislocalized proteins begin degraded in a mammalian system (PMID 24981174 and 21743475), although the factors are different.

- 1- Analysis of Pgc1 membrane association by alkaline treatment indicated that it was stably inserted in membranes both in *wt* and *doa10Δ* cells. This was further supported by the membrane flotation analysis (presented now). Given the well characterized role of Cdc48 in membrane extraction of ubiquitinated ERAD substrates, like the reviewer, we were puzzled by the lack of a phenotype in *cdc48-3* mutant cells. Therefore we decided to re-evaluate the requirement of Cdc48 function using a different, presumably more stringent, Cdc48 allele (*cdc48-6*). Inactivation of Cdc48 function in cells expressing *cdc48-6* strongly delayed the degradation of both Pgc1 and Dga1. Moreover, we detected increased levels of ubiquitinated Pgc1 associated with membranes in the *cdc48-6* cells. Together these results indicate that Cdc48 function participates in the membrane extraction of ER-localized LD proteins during ERAD. As mentioned above, we interpret that residual Cdc48 activity in *cdc48-3* is responsible for the discrepancy between the two alleles. This result suggests that, while extraction of polytopic ERAD substrates requires processive and/or multiple rounds of Cdc48 function, membrane extraction of hairpin-containing proteins like Pgc1 and Dga1 may need only a single/few Cdc48 ATPase cycle.
- 2- We expressed Pgc1 fused to the green-to-red photoconvertible fluorescent molecule tdEOS and followed the fate of ER localized EOS-Pgc1 upon induction of LD formation. We found that EOS-Pgc1 photoconverted at the ER concentrated at LDs. This is consistent with the idea that Pgc1 transits through the ER on its way to LDs and argues against the model that Pgc1 is independently targeted to ER and LDs. Together with point 1., this experiment strongly supports the model presented in the paper and we thank the reviewer for suggesting it. Finally, we discussed the similarities and differences of the protein spatial control of LD proteins and previously described degradation of mislocalized proteins.

Reviewer 3:

The manuscript by Ruggiano et al. reports a new function for the endoplasmic reticulum associated degradation (ERAD) pathway. ERAD is a protein quality control mechanism that eliminates misfolded proteins from the endoplasmic reticulum to maintain protein homeostasis. In budding yeast, it employs two major ubiquitin ligases, Hrd1p and Doa10p. The author followed up on their previous proteomic study, which identified a few lipid droplet proteins as potential substrates of the ERAD ubiquitin ligase Doa10 because the expression of these proteins was elevated in Doa10 deletion yeast cells. In this study, the authors demonstrate that this is due to increased stability. They showed that the lipid droplet (LD) protein Pgc1, Dga1, Yeh1 are unstable proteins whose degradation depends on the 26S proteasome. The degradation of these proteins also requires Doa10 and the cognate ubiquitin conjugating enzyme Ubc6p and Ubc7p, but the Cdc48p ATPase is dispensable. To characterize this process, they focused on Pgc1. They showed that Pgc1 is normally partitioned between ER and lipid droplet and only the ER pool is subject to degradation by the ERAD pathway. They further show that the long transmembrane domain of Pgc1 is both necessary and sufficient for degradation by the proteasome. The study identifies a new class of substrate for the ERAD pathway (some lipid droplet proteins mislocalized to the ER membrane, and suggests that ERAD may have a quality control function to remove mislocalized LD proteins. Overall, the conclusions are well supported by the results. I only have a few relatively minor points for the authors to consider, which I believe will make the paper a stronger candidate for EMBO.

1. The authors tested the degradation of Pgc1 in Cdc48 and Npl4 mutant yeast strains and found no effect. They mentioned that the strains were characterized previously by testing a bona fide ERAD substrate, whose degradation is strongly inhibited by mutations in Cdc48 gene. However, that was done separately. It would be more convincing if the authors can put Erg1 together with Pgc1 in the same strain and show that one is affected by not the other. I am a little bit concerned that the

negative data here might be due to some trivial experimental error because in a paper cited by the authors, a mammalian LD protein was recently demonstrated to be an ERAD substrate, but the degradation requires p97, the mammalian homolog of Cdc48.

2. In Figure 4C, the authors wish to demonstrate that oleate acid feeding, a procedure that induces LD formation may stabilize Pgc1. However, the difference is too small to be significant. It may be better to compare the stability of Pgc1 in the mutant yeast cells that do not have LD (those shown in Figure 3A before induction) and those that have the Dga1 expression induced (LD induction condition shown in Figure 3A) or perform the oleate acid feeding experiment in mutant cells that do not have any LD to begin with.

Minor points:

- 1. On page 9, the authors mentioned Erg6 as "a well-characterized LD marker protein". When I searched PubMed, I could not find many papers showing this as an LD marker. Please give the reference.*
- 2. On page 13, the authors mentioned that their study reveals "a function for the ERAD pathway that is distinct from its role in protein quality control". In a broad sense, the newly demonstrated ERAD function can still be considered as quality control as it removes LD proteins mislocalized to the ER, which is analogous to the cytosolic quality control pathway that deals with mislocalized ER membrane proteins.*
- 3. Figure 1B, what do I and U stand for? Please explain.*
- 4. Can standard deviation be calculated from two experimental datasets? It may be more appropriate to say that the error bars indicate the range of two experimental repeats. With key experiments (such as those shown in Figure 4B), it would be more convincing if the authors provides 3 repeats instead of 2, so can add p value to the graph. Some quantification graphs appear too busy. It may be better to separate the curves into 2 or more panels, so the comparison is more obvious? The FRPA experiments need to explain the number of LD analyzed in the figure legend.*

- 1- Please see our response to Reviewer 2, point 1. Also we would like to clarify that in all the experiments with *Cdc48* mutant alleles, the analysis of the control ERAD substrate Erg1 was always performed in the same cells. In all cases we used a previously described anti-Erg1 antibody to look at endogenous Erg1.
- 2- We agree with the reviewer that in the original manuscript there was only a marginal effect of the oleate feeding on the kinetics of degradation of Pgc1. Now, by performing longer oleate feeding, preventing triglyceride lipolysis or both, we detect more pronounced delays on Pgc1 degradation. We note the strength of the effect increases with the expansion of LD surface consistent with the notion that the LD pool of Pgc1 is protected from Doa10-dependent degradation. In agreement with this conclusion, the kinetics of degradation of Vma12-Ndc10C', a Doa10 substrate that does not localize to LDs, was unaffected under all the tested conditions.

Minor points:

- 1- References have been added.
- 2- We now reference and discuss our data in light of previous findings on the role of quality control pathways in degrading mislocalized proteins.
- 3- The labelling of Figure 1B has been corrected.
- 4- The number of repetitions of all experiments has now been increased and the number of repetitions is indicated. In the FRAP experiment 10 cells of each genotype were analysed. This information is included in the Material and Methods section.

Thank you for submitting your revised manuscript for our consideration. It has now been seen once more by the original referees (see comments below), and I am happy to inform you that they are broadly in favor of publication, pending satisfactory minor revision.

I would therefore like to ask you to address referee #3's remaining concern and to provide a final version of your manuscript.

REFEREE COMMENTS

Referee #1:

The authors have responded strongly and positively to the reviewers' criticisms. Two additional lines of experimentation are provided. The first is a photoconversion assay that in effect represents a pulse-chase experiment to demonstrate that Pgc1 loads into LDs from an ER pool rather than a cytosolic pool. These data are consistent with the ERAD machinery degrading Pgc1 in the ER when loading into LDs is inhibited. Second, the authors resolve what was a paradoxical result regarding the role of Cdc48 in the degradation reaction. Using a tighter allele, they now find Cdc48 is indeed required for Pgc1 degradation in the ER therefore making this a canonical ERAD reaction. The demonstration that polyubiquitinated Pgc1 accumulates in the ER of *cdc48-6* mutants supports that conclusion.

The technical quality of the work is high, it is clearly written, and most definitely deserves publication. This reviewer remains on the fence as to whether their refined definition of spatially controlled degradation as it relates to ERAD and LD cargo is of sufficient conceptual novelty to merit publication in EMBO J. This reviewer remains of the opinion that this quality work is better suited for a more specialized journal.

Referee #2:

The authors have addressed the two main issues I had raised with additional experiments and discussion. The new results are convincing, and I am happy to recommend publication of this interesting study in EMBO journal.

Referee #3:

The authors have addressed my criticisms adequately. There is only one minor issue remaining. In the previous version, the authors used the *Cdc48-3* allele and a *Npl4* mutant strain. They found that inactivation of Cdc48 or its co-factor did not affect ERAD of *pgc1*, but now in the revised version, they used a different Cdc48 mutant strain to reach the opposite conclusion. They argue that the two different Cdc48 mutant strains may not cause similar degree of lost-of-function, but the readers are left wondering about the cause of this difference. It would be nice if the authors could sequence the Cdc48 gene in these two strains, which may allow them to provide a plausible explanation for such a difference. At minimum, the authors should provide the source and a reference for the *Cdc48-6* allele.

2nd Revision - authors' response

01 June 2016

We are submitting a final version of the manuscript in which we addressed the minor issue raised by reviewer #3. Previous studies describing the phenotypes of the *CDC48* alleles used in our work are now referenced. Moreover, we followed the reviewer's suggestion to sequence the two alleles. The data shows that *cdc48-3* contains mutations only in the D1 ATPase domain (P257L and R387K), whereas in agreement with its tighter phenotype, *cdc48-6* contains mutations in both D1 (P257L) and D2 (A540T) ATPase domains. These findings support our conclusion that membrane extraction of hairpin-containing LD proteins (like Pgc1 and Dgal) may require only residual Cdc48 activity. These results are now included in the final manuscript.

Accepted

02 June 2016

Thank you for submitting your revised manuscript to us. I appreciate the additional insight into the *cdc48* alleles offered and I am happy to inform you that your manuscript has been accepted for publication in the EMBO Journal.
Congratulations!

Corresponding Author Name: Pedro Carvalho

Journal Submitted to: The EMBO Journal

Manuscript Number: EMBOJ-2015-93106R